# miRNA Signatures in Alveolar Macrophages Related to Cigarette Smoke: Assessment and Bioinformatics Analysis

**DOI:** 10.3390/ijms26031277

**Published:** 2025-02-01

**Authors:** Davida Mirra, Renata Esposito, Giuseppe Spaziano, Concetta Rafaniello, Francesca Panico, Antonio Squillante, Maddalena Falciani, Diana Marisol Abrego-Guandique, Eleonora Caiazzo, Luca Gallelli, Erika Cione, Bruno D’Agostino

**Affiliations:** 1Department of Environmental Biological and Pharmaceutical Sciences and Technologies, University of Campania “Luigi Vanvitelli”, 81100 Caserta, Italy; davida.mirra@unicampania.it (D.M.); renata.esposito@unicampania.it (R.E.); eleonora.caiazzo@studenti.unicampania.it (E.C.); bruno.dagostino@unicampania.it (B.D.); 2Campania Regional Centre for Pharmacovigilance and Pharmacoepidemiology, 80138 Naples, Italy; concetta.rafaniello@unicampania.it; 3Section of Pharmacology “L. Donatelli”, Department of Experimental Medicine, University of Campania “Luigi Vanvitelli”, 80138 Naples, Italy; 4Science of Health Department, School of Medicine, University of Catanzaro, 88100 Catanzaro, Italy; francesca.panico@studenti.unicz.it (F.P.); dianamarisol.abregoguandique@unicz.it (D.M.A.-G.); 5Department of Medicine, University of Salerno, 84100 Salerno, Italy; a.squillante92@gmail.com; 6Pulmonary and Critical Care Medicine, Ospedale Scarlato, 84018 Scafati, Italy; m.falciani@aslsalerno.it; 7Clinical Pharmacology and Pharmacovigilance Unit, Department of Health Sciences, Mater Domini Hospital, University of “Magna Graecia”, 88100 Catanzaro, Italy; gallelli@unicz.it; 8Department of Pharmacy, Health and Nutritional Sciences, University of Calabria, 87036 Rende, Italy; erika.cione@unical.it

**Keywords:** microRNAs, alveolar macrophages, cigarette smoke, COPD, lung cancer

## Abstract

Cigarette smoke (CS) is a driver of many respiratory diseases, including chronic obstructive pulmonary disease (COPD) and non-small cell lung cancer (NSCLC). Tobacco causes oxidative stress, impaired phagocytosis of alveolar macrophages (AMs), and alterations in gene expression in the lungs of smokers. MicroRNAs (miRNAs) are small non-coding RNAs that influence several regulatory pathways. Previously, we monitored the expressions of hsa-miR-223-5p, 16-5p, 20a-5p, -17-5p, 34a-5p, and 106a-5p in AMs derived from the bronchoalveolar lavage (BAL) of subjects with NSCLC, COPD, and smoker and non-smoker control groups. Here, we investigated the capability of CS conditionate media to modulate the abovementioned miRNAs in primary AMs obtained in the same 43 sex-matched subjects. The expressions of has-miR-34a-5p, 17-5p, 16-5p, 106a-5p, 223-5p, and 20a-5p were assessed before and after in vitro CS exposure by RT-PCR. In addition, a comprehensive bioinformatic analysis of miRNAs KEGGS and PPI linked to inflammation was performed. Distinct and common miRNA expression profiles were identified in response to CS, suggesting their possible role in smoking-related diseases. It is worth noting that, following exposure to CS, the expression levels of hsa-miR-34a-5p and 17-5p in both smokers and non-smokers, 106a-5p in non-smokers, and 20a-5p in smokers, shifted towards those found in individuals with COPD, suggesting them as a risk factor in developing this lung condition. Moreover, CS-focused sub-analysis identified miRNA which exhibited CS-dependent pattern and modulated mRNA involved in the immune system or AMs property regulation. In conclusion, our study uncovered miRNA signatures in AMs exposed to CS, indicating that CS might modify epigenetic patterns that contribute to macrophage activation and lung disease onset and progression.

## 1. Introduction

Cigarette smoke (CS) is the leading cause of preventable deaths worldwide and is commonly considered a major driver of many respiratory diseases, including non-small cells lung cancer (NSCLC) and chronic obstructive pulmonary disease (COPD) [1,2,3]. COPD is an heterogenous disease characterized by the progressive deterioration of lung function over time and is generally associated with lung inflammation triggered by harmful particles or gases [4,5,6,7,8,9,10]. COPD and lung cancer, beyond a common etiology, are closely linked conditions, and patients with COPD have twice the risk of cancer diagnosis [11,12,13,14,15,16]. Tobacco promotes oxidative stress, inflammation, and genetic and molecular impairments, which may increase the chance of mutations and lung carcinogenesis [17,18,19,20]. Moreover, it is becoming increasingly evident that the development of COPD or NSCLC phenotypes in response to harmful agents is regulated by both the innate and adaptive immune systems [21,22,23]. Alveolar macrophages (AMs) are essential effector cells in the innate lung immune system, as they recruit other immune cells, phagocytize inhaled environmental particles, and release pro-inflammatory mediators [24,25,26,27,28]. Smoking causes AM impairment in phagocytosis and responses to pathogens, compromising their protection from noxious agents [29,30,31]. Importantly, AM gene expression can be altered in response to environmental exposure, leading to epigenetic changes such as DNA methylation, covalent histone modifications, and microRNAs (miRNA) expression [32,33]. miRNAs are small non-coding endogenous RNA molecules capable of modulating mRNA degradation and suppression, thereby coordinating gene expression under pathological conditions including tumors and inflammatory lung diseases [34,35,36,37,38]. Recently, our group monitored the expression of the signature hsa-miR-34a-5p, 17-5p, 16-5p, 106a-5p, 223-5p, and 20a-5p in bronchoalveolar lavage (BAL) of subjects with NSCLC, COPD, and smoking or non-smoking controls, suggesting their potential role as an index of the smoking-related disease microenvironment [39]. Notably, all selected miRNAs have been shown to influence processes related to inflammation, carcinogenesis, or immunity, which are closely linked to CS [40,41,42,43,44,45]. Several authors have attempted to describe a miRNA signature that correlates with CS-associated clinical phenotypes and lung inflammation. Willinger et al. profiled 283 miRNAs in whole blood and revealed that five of the six CS-related miRNAs were associated with C-reactive protein serum levels or interleukin-6 and has-miR-1180 with pulmonary function [46]. Karabegović et al. reported an association between CS and 591 well-expressed miRNAs in plasma samples, identifying 41 smoking-associated miRNAs, 8 of which were associated with a higher incidence of lung cancer (including miR-146b-5p, miR-6769b-3p, miR-1915-3p, miR-6085, miR-10a-5p, miR-100-5p, miR-149-3p, and let-7c-5p). The same authors, exploring a possible association between CS and miRNA for the predicted target genes, identified 407 predicted target genes, of which BRWD1, CADM1, and EED were already associated with smoking in a genome-wide association study [47,48]. However, little is known about the effect of active smoking on the expression levels of miRNAs in primary AMs and how it affects the identification of miRNA signature of pulmonary conditions. Therefore, we evaluated the levels of the above-mentioned-miRNAs in AMs isolated from BAL of smoking or non-smoking controls and patients with COPD or NSCLC before and after CS exposure.

## 2. Results

### 2.1. CS Effects on TH-P1 Metabolic Activity

To obtain a CS treatment that can trigger a cellular response while avoiding excessive AMs damage, we used TH-P1 macrophage cell line. The treatment of TH-P1 cells with CS for 24h significantly affects metabolic activity, which peaked at 10% CS (*p* < 0.0001) (Figure 1A). No significant differences were observed in cell morphology in any CS treatment (Figure 1B). MTT assays in AMs obtained from three healthy non-smoking donors confirmed that the maximum reduction in metabolic activity is reached after 10% CS for 24h in all samples, corroborating what was observed in TH-P1 cells. Therefore, 24h 10% CS was used to perform AM exposure.

### 2.2. miRNA Expression Levels

hsa-miR-34a-5p, 17-5p, and 16-5p

We assessed miRNA signatures in each pathological condition (NSCLC and COPD), smoking habit (HS), and control group (HNS) before and after exposure to 10% CS for 24 h. As we previously reported, lung conditions like COPD or NSCLC as well as exposure during lifetime to CS significantly affected miRNAs expression determining; therefore, different results depending on the type of subject from which AMs were obtained [40]. Indeed, following stimulation with 10% CS for 24 h, we observed a significant increase in hsa-miR-34a-5p (*p* < 0.01), 17-5p (*p* < 0.0001), and 16-5p (*p* < 0.01) expression in AMs obtained from HNS group (Figure 2A–C). Interestingly, acute in vitro CS stimulation also led to significant positive modulation of hsa-miR-34a-5p (*p* < 0.0001), 17-5p (*p* < 0.0001), and 16-5p (*p* < 0.0001) expressions in AMs from HS (Figure 2A–C). In contrast, the CS stimulation of COPD and NSCLC AMs did not affect hsa-miR expression (Figure 2A–C). This could suggest that acute CS stimulation is sufficient in exclusively affecting hsa-miR expression in AMs from subjects without pre-existing lung diseases.

hsa-miR-106a-5p

Following in vitro exposure to 10% CS for 24 h, we observed a significant increase in hsa-miR-106a-5p expression in HNS AMs (*p* < 0.05) and in AMs from the COPD group (*p* < 0.001) (Figure 3A). Interestingly, CS led to the opposite trend in AMs from HS (*p* < 0.05), while it did not affect hsa-miR-106a-5p expression in AMs obtained from patients with NSCLC (Figure 3A).

hsa-miR-223-5p and 20a-5p

CS significantly decreased hsa-miR-223-5p (*p* < 0.001) and 20a-5p (*p* < 0.001) expression in HNS AMs (Figure 3B,C). In contrast, in vitro CS stimulation in COPD AMs induced both hsa-miR-223-5p (*p* < 0.0001) and 20a-5p (*p* < 0.05) upregulation (Figure 3B,C). Moreover, CS affected only hsa-miR-20a-5p expression in AMs from HS, leading to a significant positive modulation (*p* < 0.0001) (Figure 3), but it did not further impact hsa-miR expression in AMs from individuals with NSCLC (Figure 3B,C).

### 2.3. In Silico Identification of Target mRNAs

The relationship between miRNAs and lung response to CS was assessed by in silico analysis of sequence similarity between miRNAs and different mRNAs. The results, reported in Figure 4 and Figure 5, showed that the miRNA may be involved in the regulation of 740 genes involved in inflammation pathways, such as BCL2, MTOR, MCL1, TGFBR, SMAD, or VEGF, among others (for comprehensive analysis, see Appendix A).

### 2.4. CS-Focused Sub-Analysis

COPD and NSCLC may represent two confounding variables which make data interpretation challenging. Moreover, the HS group consisted of 11 subjects, of which 3 were former smokers and 8 were current smokers. Therefore, we performed a sub-analysis focused on healthy subjects to highlight the effects of active CS on miRNA expression levels. Specifically, non-smokers (n = 9) in the HNS group exposed or not to in vitro CS, and current smokers (n = 8) in the HNS group were included. As shown in Figure 6, the increase in hsa-miR-106a-5p and 17-5p levels in HNS CS-exposed AMs (*p* < 0.0001 and *p* < 0.001, respectively) were comparable to the levels observed in current HS (*p* < 0.0001 and *p* < 0.05, respectively). Similarly, the decrease in hsa-miR-20a-5p and 22-5p levels in HNS exposed to CS (*p* < 0.01 and *p* < 0.001, respectively) reflects those noted in current HS (*p* < 0.001 and *p* < 0.01, respectively). No comparable trends were observed in the expression of hsa-miR-34a-5p and 16-5p.

Based on these results, we performed further bioinformatics analysis focused on has-miR-106a-5p, 17-5p, 20a-5p, and 223-5p 34a-5p, which exhibit CS-dependent modulation. KEGG analysis of the target genes showed that cellular senescence exhibited the highest enrichment score (−log 10 (*p* value) ≈ 10) (Figure 7). Similarly, the p53 signaling pathway, a key regulator of the cell cycle and apoptosis, showed a high enrichment score and significance (−log 10 (*p*-value) ≈ 8) (Figure 7). Appendix A show the gene ontology (GO) analysis of the target genes and PPI network, respectively. MCODE allowed us to obtain a subnetwork with 35 nodes and 390 edges (A) and one with 25 nodes and 134 edges (B) (Figure 8). CREB1 represented as the central seed node, interacts with multiple highly connected proteins, such as HIF1A, TP53, MYCN, ATM, and BCL2 from the first subnetwork and E2F1 (seed node), STAT3, MYC, MAPK1, and BRCA2, from the second network, suggesting that these nodes may play key roles in the networks. Finally, Reactome Pathway Enrichment identifies signal transduction and TGFB family member signaling as the most represented and enriched pathways, respectively (Figure 9) (for comprehensive analysis, see Appendix A).

## 3. Discussion

In this study, we assessed the effect of in vitro CS exposure on the regulation of miRNA expression in AMs, the main participants in the development of smoking-related conditions such as COPD and NSCLC. Notably, tobacco promotes oxidative stress and systemic and local inflammation in the lungs of smokers, leading to innate and adaptive immune system impairments [18,19,20]. In addition, CS leads to genetic and molecular impairments, which may increase the chance of mutations and lung carcinogenesis [21]. In our previous study, we reported that has-miR-34a-5p, 17-5p, 16-5p, 223-5p, 20a-5p, and 106a-5p expression profiles were dysregulated in NSCLC, COPD, and smoking or non-smoking controls, suggesting their possible role as an index of smoking-associated conditions [39]. To further investigate the effect of active smoking on the expression levels of these miRNAs, we analyzed the changes in their expression before and after in vitro CS exposure in the above-mentioned groups. This profiling was carried out in AMs recovered from BAL, a precious biological sample that is highly representative of the pulmonary microenvironment [49]. First, we identified that the AMs of non-smokers in vitro stimulated with 10% CS for 24h results in two specific trends, leading to hsa-miR-34a-5p, 17-5p, 16-5p, and 106a-5p upregulation and the negative modulation of hsa-miR-223-5p and 20a-5p levels. In contrast, in vitro exposure of AMs to CS leads to different modulations of miRNAs according to smoking habits or COPD and NSCLC presence. Apoptosis is a hallmark of parenchyma impairment and COPD development and can be triggered by CS in several cell types, including macrophages [50]. In this context, Long et al. reported that Notch-1 receptor protein, a transmembrane receptor implicated in cell apoptosis, is lower expressed in primary lung microvascular endothelial cells (HPMECs) treated with CS, with the upregulation of hsa-miR-34a-5p [51]. Furthermore, in our previous study, we reported a significant positive modulation of hsa-miR-34a-5p in the tissue and AMs of COPD subjects compared to healthy never-smoker controls [52]. Consistent with these data, we observed that, after in vitro exposure to CS, a significant positive modulation of hsa-miR-34a-5p in the AMs of healthy individuals occurred at the COPD level, providing evidence of the role of CS in COPD-like dysregulation. Interestingly, in vitro CS exposure equally affected hsa-miR-34a-5p ex-pression in AMs obtained from HS suggesting the potent effect of acute CS stimulation. Leukocyte signal regulatory protein-α (SIRPα), a member of the immunoglobulin superfamily, modulates many aspects of the inflammatory response, including immune cell activation, chemotaxis, and phagocytosis [53]. In this regard, Zhu et al. showed that the upregulation of hsa-miR-17-5p by lipopolysaccharide (LPS) in macrophages correlates to SIRPα reduction and AM activation [54]. This was consistent with our finding of higher hsa-miR-17-5p levels in AMs of non-smokers and smokers following acute in vitro CS treatment, supporting the importance of CS in the mechanisms underlying AM impairment in lung diseases. One of the most important features of host defense against harmful is the activation of TLR immune receptors and the release of a variety of toxic products, including reactive oxygen species (ROS) such as NO, hydrogen peroxide, and superoxide anions [55,56]. Moon et al. reported that bacterial LPS enhanced the level of has-miR-16-5p in bone marrow-derived macrophages, resulting in decreased phagocytosis and the generation of mitochondrial ROS [57]. Accordingly, our findings showed a positive modulation of has-miR-16-5p in the AMs of non-smokers and smokers following CS treatment, suggesting the ability of CS exposure to modulate the lung inflammatory response. Although a few studies investigated the hsa-miR-106a-5p expression patterns associated with CS and chronic lung diseases, Liu et al. reported that it dramatically inhibited the activation of autophagy induced by M. tuberculosis in human TH-P1 macrophages [58]. Moreover, Sharma et al. reported that hsa-miR-106a-5p negatively regulates IL-10 expression with an increase in pro-inflammatory cytokines in in vitro and in vivo model of airway inflammation [59]. This was in line with our findings of increased has-miR-106a-5p in AMs of both non-smokers and COPD subjects following in vitro CS exposure. However, the regulatory effects of has-miR-106a-5p in CS-related diseases are not fully understood, making its role controversial, as suggested by its reduction in AMs from smokers after in vitro CS treatment. Our findings highlight that the in vitro CS stimulation of AMs obtained from non-smokers results in the negative modulation of hsa-miR-223-5p and 20a-5p levels. Several authors described hsa-miR-223-5p’s role in macrophage differentiation, neutrophil recruitment, and pro-inflammatory responses, which are key features of lung inflammation and remodeling [60]. Interestingly, in non-smokers AMs, CS led to the modulation of hsa-miR-223-5p to levels comparable to those observed in individuals with smoking-related conditions. Consistent with our data, Schembri et al. reported lower has-miR-223-5p levels in bronchial epithelial cells from current smokers than in those from non-smokers [61]. Furthermore, it is important to point out that in COPD AMs, the expression of this miRNA increased following exposure to CS, indicating a unique function for acute CS in COPD microenvironment. In fact, acute CS exposure can induce chemotactic factors in the lungs, stimulate AMs, and lead to neutrophil influx, which can require at least six months to completely normalize [62]. In this context, Roffel et al. detected higher levels of has-miR-223-5p in the lung tissue of COPD patients, assuming that it could be associated with impaired lung function and higher neutrophil counts [63]. As for hsa-miR-223-5p, CS led to hsa-miR-20a-5p downregulation in AMs obtained from non-smokers. Importantly, our data showed that smokers and patients with COPD shared increased levels of this miRNA after exposure to CS. In particular, exposure to CS in smokers increases levels towards those reported in COPD, highlighting the close link between CS and the development of a COPD-like phenotype. hsa-miR-20a-5p has been shown to regulate AMs inflammatory responses by targeting SIRPα [53]. Moreover, Liu et al. reported that higher hsa-miR-20a-5p levels in children with pneumonia and in lung cells exposed to LPS are linked to the NF-κB signaling pathway [64]. However, given its role in controlling different cellular networks, the regulatory effects of CS on its expression cannot be generalized, making a more in-depth analysis necessary to explain our results in AMs from never-smokers [65]. In vitro CS exposure did not influence the expression of any miRNAs in AMs from subjects with NSCLC. In our previous study, the same trend was seen for the programmed death-ligand 1 (PD-L1) mRNA expression. Indeed, we reported that after CS exposure, PD-L1 mRNA expression was increased in AMs derived from non-smoking subjects but not in NSCLC patients, suggesting an overwhelming effect of cancer on acute CS exposure [66]. Therefore, we performed a sub-analysis focused on non-smokers (HNS group) and current healthy smokers (HS group) to highlight the direct effect of CS on miRNA expression levels. The results showed a comparable trend in has-miR-106a-5p, 17-5p, 20a-5p, and 223-5p 34a-5p expression levels between CS-stimulated HNS AMS and current HS, supporting the hypothesis of a direct causal effect of CS on miRNA modulation. However, it is important to note that the intensity of the reaction to immunogenic antigens produced in response to CS varies across a wide range of disease manifestations, highlighting the crucial role of immune responses in regulating the development of distinct phenotypes in response to CS [67,68]. Therefore, the dysregulation of miRNAs could reflect phenotype switching or the onset of different lung manifestations, underlining the prominent, but not exclusive, role of CS. Indeed, miRNA expression profiles can be influenced by other environmental factors, which can further modulate the correlation between miRNAs expression and mRNA targets in response to CS [69]. Finally, the bioinformatics results revealed that the miRNAs analyzed may potentially be involved in the regulation of 740 inflammatory driver genes and protein networks linked to apoptosis or cytokines production [43,70,71,72,73,74,75]. Specifically, the sub-analysis focused on has-miR-106a-5p, 17-5p, 20a-5p, and 223-5p 34a-5p, which exhibit CS-dependent modulation, highlights miRNA involvement in key signaling pathways such as cellular senescence or in modulation of p53 and MYC gene, which are frequently implicated in processes such as inflammation, cellular aging or tumor onset and progression [76,77]. Moreover, the presence of TGFβ, FoxO, and SMAD signaling pathways suggests potential roles in regulating cellular responses to environmental stress such as CS [78,79,80]. Since several authors experimentally confirmed all miRNA-regulated genes, miRNAs may modulate these targets combined or individually, affecting different hallmarks of the lung response to CS. A potential limitation of our study was the small sample size used for miRNA analysis, which did not allow for sub-analysis on the severity of COPD. However, our data could be of clinical relevance and guide future studies involving larger populations, allowing for a better understanding of the networks involved in the pathogenesis of smoking-related diseases.

## 4. Conclusions

This study, albeit preliminary, indicates that CS could be a driver of epigenetic changes via microRNAs that contribute to the onset of various lung diseases. It is worth noting that, following exposure to CS, the expression levels of hsa-miR-34a-5p and 17-5p in both smokers and non-smokers, 106a-5p in non-smokers, and 20a-5p in smokers shifted to those found in individuals with COPD, likening them to a risk factor in the development of this lung condition and potential biomarkers or therapeutic targets. 

## 5. Materials and Methods

### 5.1. Study Population and Bronchoalveolar Lavage

This study is part of a clinical study recorded at clinicaltrials.gov (NCT04654104). We enrolled 43 individuals at the “Mater Domini” Hospital in Catanzaro, Italy, who underwent bronchoscopy and BAL for suspected pulmonary neoplasia [81]. Informed consent was obtained from all individual participants included in this study (for details see Appendix A). Based on the clinical data, we divided the enrolled subjects into the following groups: (1) healthy non-smoking controls (“HNS”; n = 9); (2) healthy smoking controls (“HS”; n = 11); (3) smokers with Global Initiative for Obstructive Lung Diseases (GOLD) stage 1–4 (“COPD,” n =11); (4) individuals with non-small cell lung cancer (“NSCLC”; n = 12). The main clinical and pathological characteristics of the cohorts are reported in our previous study and are available in the online version at https://doi.org/10.3390/biomedicines12051050 (accessed on 4 October 2024). BAL was obtained as previously reported and used to obtain primary AMs [39] (for details, see Appendix A).

### 5.2. Preparation of CS Extract and Cell Culture Procedures

The CS extract, prepared as previously described, was used on macrophages from acute monocytic leukemia (TH-P1) as a pilot model to establish the exact dose (2%, 5%, or 10%) able to induce metabolic impairment at 24 h using the Thiazolyl Blue Tetrazolium Bromide solution (MTT) assay (for details, see Appendix A). Cell morphology was assessed by QUICK-DIFF staining.

The BAL cell pellets were suspended in RPMI-1640 medium supplemented with 10% FBS, 2 mM L-glutamine, 200 U/mL penicillin, and 200 mg/mL streptomycin (for details see Appendix A). To confirm the results obtained in TH-P1 cells, we performed an additional MTT assay in AMs obtained from three healthy non smokers doners. Then, the purified primary AMs were then exposed to 10% CS for 24 h, based on previous treatments results.

### 5.3. Biochemistry Assays and Real-Time PCR (RT-PCR) 

The extraction of miRNAs in AMs obtained from BAL was carried out through the miRNeasy mini kit, as previously described [39], and the expression levels of has-miR-34a-5p, 17-5p, 16-5p, 106a-5p 223-5p, and 20a-5p were determined using TaqMan™ Advanced miRNA Assay RT-PCR (Waltham, MA, USA) and applying the 2ˆ^(−DDCt)^ method (for details, see Appendix A).

### 5.4. Bioinformatics Analysis

The mRNA targets of has-miR-34a-5p, 17-5p, 16-5p, 106a-5p 223-5p, and 20a-5p linked to inflammation or AM properties were analyzed using DIANA Tools and miRpath v3. Then, for the CS-focused sub-analysis, the multimiR package that integrates miRecords, miRTarBase, and Tarbase [82] was used as described [83]. Moreover, Clusterprofiler R packages were used to perform KEGG pathway enrichment analyses, and the Gene Ontology (GO) enrichment analysis was obtained using the Enrichplot. Finally, the STRING website was utilized to identify the protein–protein interaction (PPI) and to generate Reactome Pathways Enrichment [83]. Sub-networks were built using the MCODE Cytoscape plugin [84]. The most relevant nodes were selected based on biological relevance and metrics such as betweenness centrality, closeness centrality, and topological coefficient. Then, consideration was given to interactions with a medium confidence score > 0.4 (for details, see Appendix A).

### 5.5. Statistical Analysis

Unless specified, all data are expressed as mean ± standard deviation (SD). Differences were considered statistically significant at *p* < 0.05 (for details, see Appendix A).

## Figures and Tables

**Figure 1 ijms-26-01277-f001:**
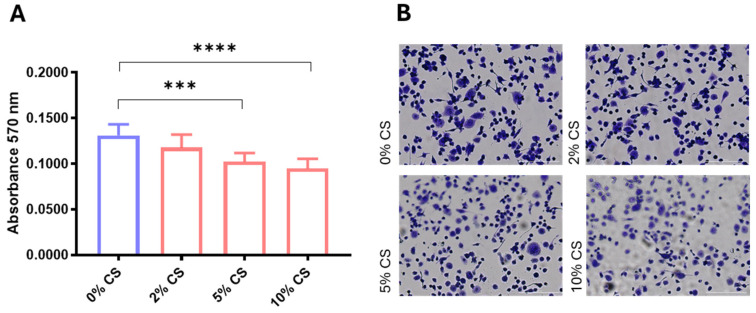
TH-P1 metabolic activity (**A**) and morphology (**B**) after CS exposure. TH-P1 cells were treated with CS at 24 h for the indicated concentration, and MTT assay and QUICK-DIFF staining were performed to assess cell metabolism and morphology, respectively. Metabolic activity is shown as absorbance at 570 nm. All samples were run in triplicate, and results are shown as means ± SD. The statistical tests used in these analyses were one-way analysis of variance followed by Dunnet Multiple-Comparison Test. *** *p* < 0.001, **** *p* < 0.0001.

**Figure 2 ijms-26-01277-f002:**
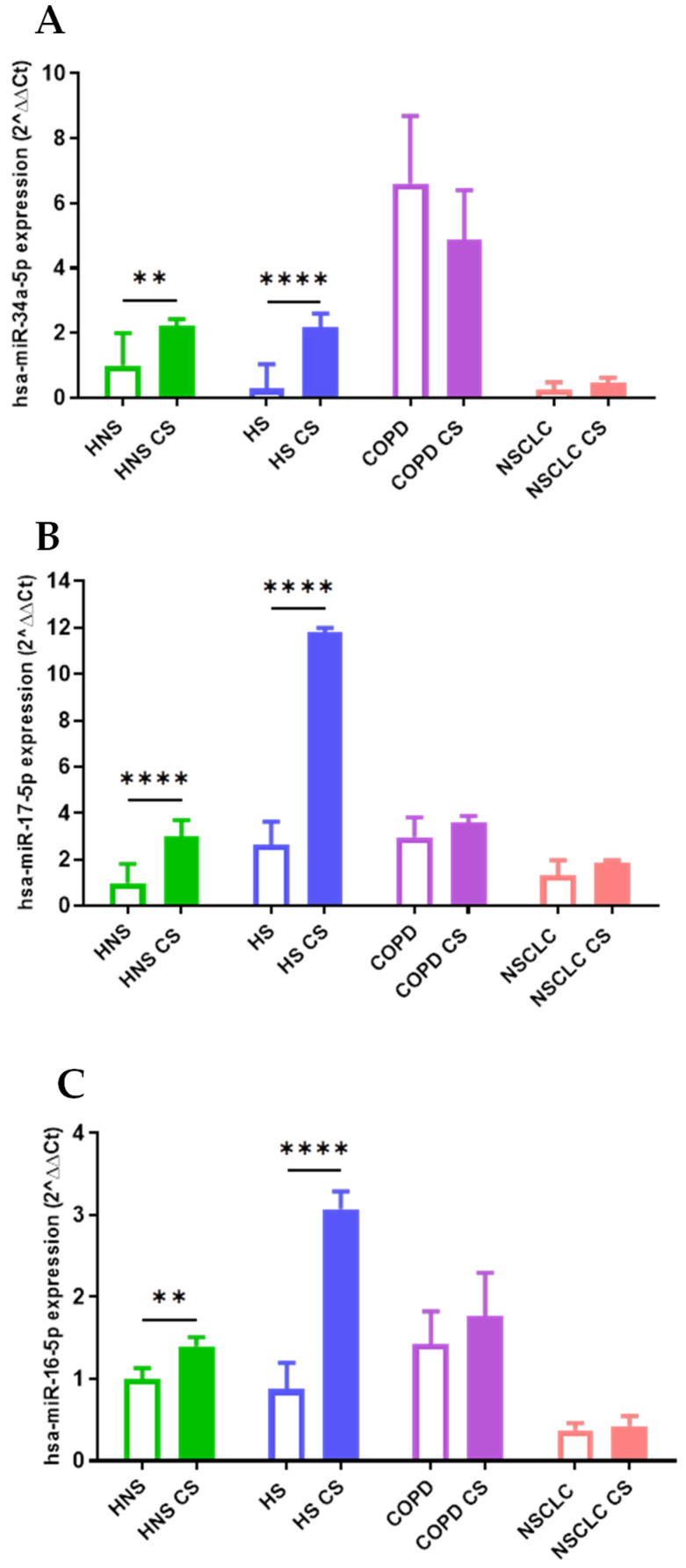
Analysis of hsa-miR-34a-5p (**A**), hsa-miR-17-5p (**B**), and hsa-miR-16-5p (**C**) AM expression levels in healthy non-smoking control (“HNS”; biological replicates n = 9), healthy smoking control (“HS”; biological replicates n = 11), smokers with Global Initiative for Obstructive Lung Diseases (GOLD) stage 1–4 (“COPD”; biological replicates n = 11), and non-small cell lung cancer (“NSCLC”; biological replicates n = 12) before and after 10% CS at 24 h. AMs were treated with CS at 10% for 24 h and the expression of hsa-miRs were assessed by real-time RT-PCR. All samples were run in triplicate, and results are shown as means ± SD. To assess the differences in miRNAs expression between AMs exposed or not to CS in each group, the one-way analysis of variance was used followed by Tukey Multiple-Comparison Test with a single pooled variance. ** *p* < 0.01, **** *p* < 0.0001.

**Figure 3 ijms-26-01277-f003:**
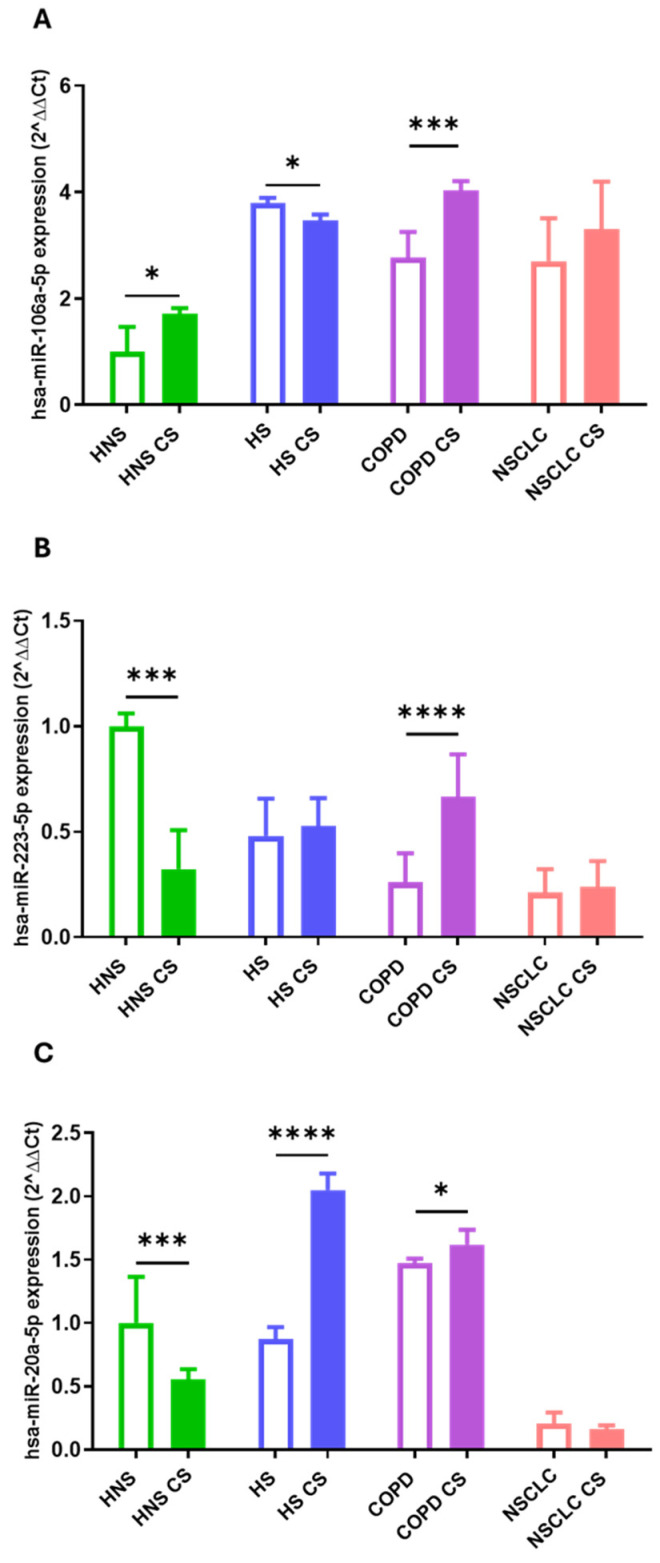
Analysis of hsa-miR-106a-5p (**A**), hsa-miR-223-5p (**B**), and hsa-miR-20a-5p (**C**) AMs expression levels in healthy non-smoking control (“HNS”; biological replicates n = 9), healthy smoking control (“HS”; biological replicates n = 11), smokers with Global Initiative for Obstructive Lung Diseases (GOLD) stage 1–4 (“COPD”; biological replicates n = 11), and non-small cell lung cancer (“NSCLC”; biological replicates n = 12) before and after 10% CS at 24 h. AMs were treated with CS at 10% for 24 h, and the expression of hsa-miRs were assessed by real-time RT-PCR. All samples were run in triplicate, and results are shown as means ± SD. To assess the differences in miRNAs expression between AMs exposed or not to CS in each group, the one-way analysis of variance was used followed by Tukey Multiple-Comparison Test with a single pooled variance. * *p* < 0.1, *** *p* < 0.001, **** *p* < 0.0001.

**Figure 4 ijms-26-01277-f004:**
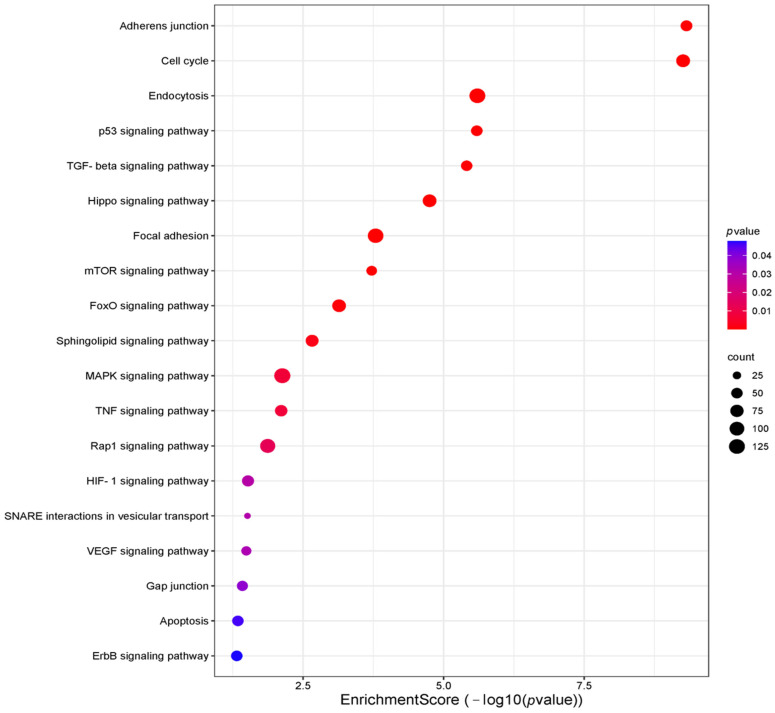
KEGG analysis of target genes. The color of each dot represents the *p* value of each term involved in the analysis. The size of each dot represents the counts of overlapped genes between the input genes and the total gene list on KEGG pathway.

**Figure 5 ijms-26-01277-f005:**
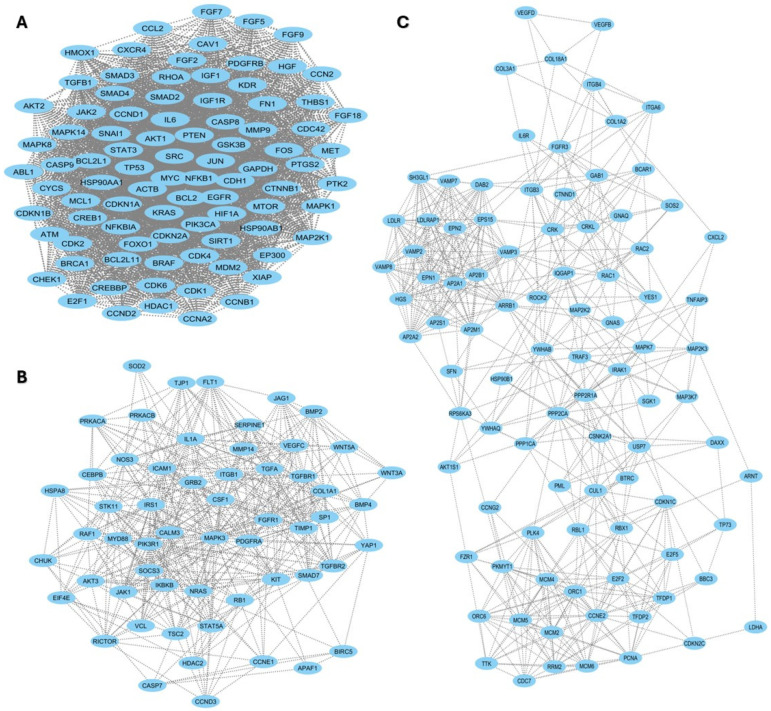
Protein–protein interaction networks. Sub-network protein–protein interaction with TGFB1, PTEN, SMAD4, MCL1, FOXO1, NFKB1, IL6, mTOR, and SIRT1 (**A**). Sub-network with VEGFC, ILA, TGFBR2, and SMAD7 (**B**). Sub-network with IL6R, VEGFB, and VEGFD (**C**). Each node is a protein, and an edge is an interaction between two proteins.

**Figure 6 ijms-26-01277-f006:**
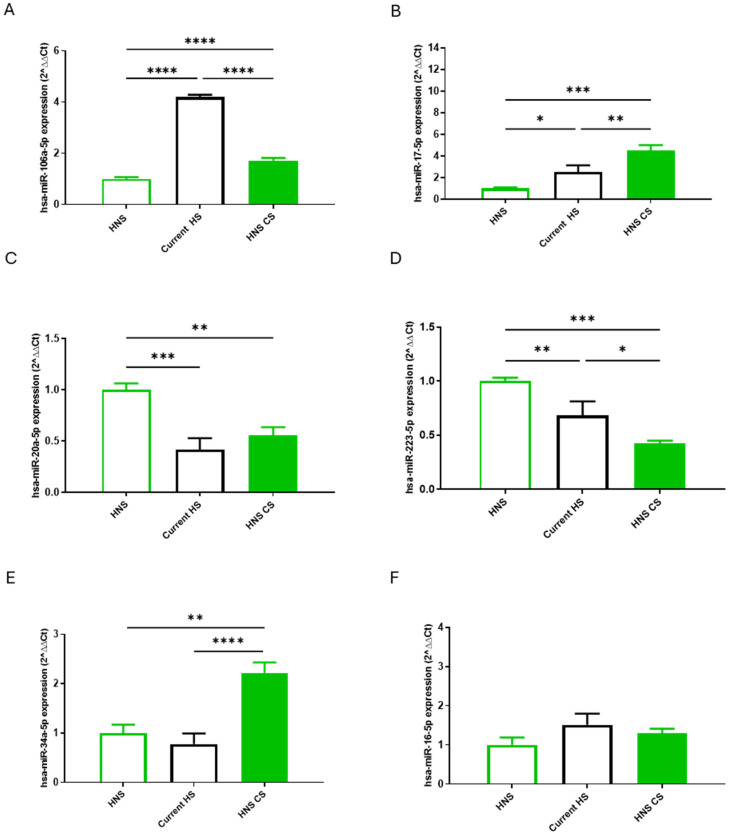
Analysis of hsa-miR-106a-5p (**A**), hsa-miR-17-5p (**B**), hsa-miR-20a-5p (**C**), hsa-miR-223-5p (**D**), hsa-miR-34a-5p (**E**) and hsa-miR-16-5p (**F**) AM expression levels in healthy never-smoker control (“HNS”; biological replicates n = 9), before and after 10% CS at 24 h, and healthy current smokers’ (“HS”; biological replicates n = 8). AMs were treated with CS at 10% for 24 h and the expression of hsa-miRs were assessed by real-time RT-PCR. All samples were run in triplicate, and results are shown as means ± SD. To assess the differences in miRNAs expression between AMs exposed or not to CS in each group, the one-way analysis of variance was used followed by Tukey Multiple-Comparison Test with a single pooled variance. * *p* < 0.05, ** *p* < 0.01, *** *p* < 0.001, **** *p* < 0.0001.

**Figure 7 ijms-26-01277-f007:**
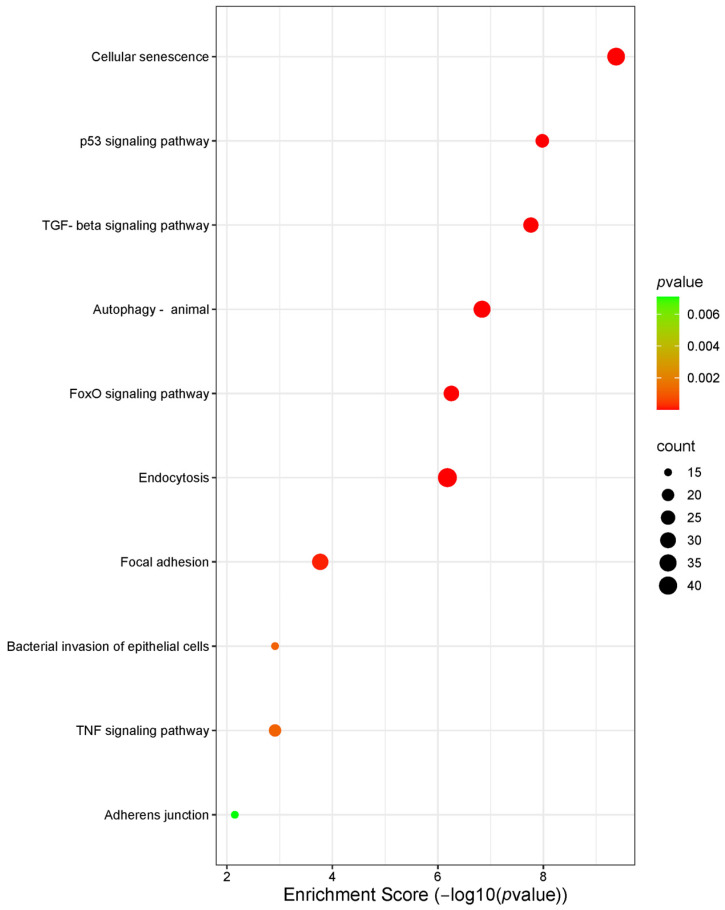
KEGG analysis of the key pathways in which target genes are involved. Thecolor of each dot represents the *p*-value of each term involved in the analysis. The size of each dot represents the counts of overlapped genes between the input genes and the total gene list on KEGG pathway.

**Figure 8 ijms-26-01277-f008:**
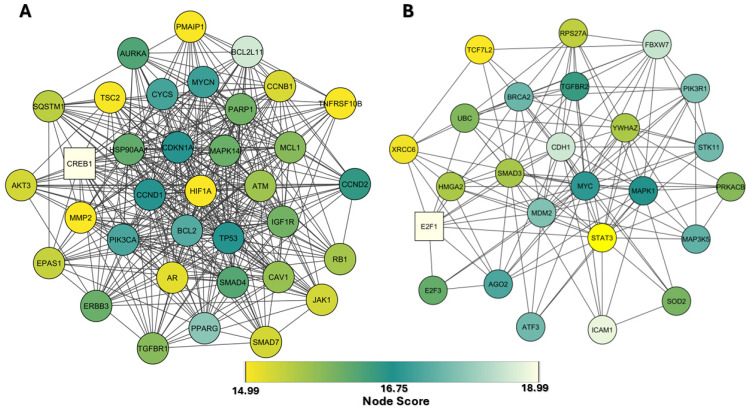
Network of the sub analysis by MCODE. (**A**) Sub-network with 35 nodes and 390 edges. (**B**) Sub-network with 25 nodes and 134 edges. The size of nodes clarifies the MCODE score, and node color clarifies betweenness.

**Figure 9 ijms-26-01277-f009:**
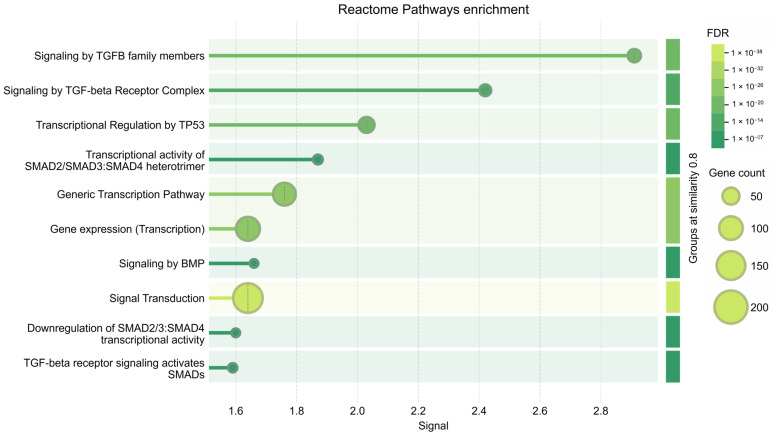
Reactome Pathways Enrichment. The color of each dot represents the FDR (false discovery rate) of each term involved in the analysis. The size of each dot represents the counts of genes.

## Data Availability

The original contributions presented in this study are included in the article/Appendix A. Further inquiries can be directed to the corresponding author.

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
