# Peer review of "miRNA Signatures in Alveolar Macrophages Related to Cigarette Smoke: Assessment and Bioinformatics Analysis"

_ijms, 2025, doi:10.3390/ijms26031277_

Round 1
Reviewer 1 Report
Comments and Suggestions for Authors
In this study, the authors evaluated the impact of cigarette smoke (CS) exposure on the expression of some miRNA and the characterization of the specific effect on the transcriptome. The effect of CS exposure was measured by in vitro experiments with AM from patients with different lung diseases (COPD or lung cancer) and by comparing the response among these different categories of patients. Although the data are interesting, the interpretation of the results is altered by the lack of controls and of the integration of some data.
Major comments
1. Since previous reports identified the modulation of others miRNA by CS exposure and/or COPD, it seems important to integrate these data at least for the in silico identification of miRNA targets.
2. The authors evaluated the impact of CS exposure on cell metabolism (by MTT assay) on THP. The decrease in MTT oxidation might be related to cytotoxicity. This might be measured by a specific test. Moreover, this experiment is used in order to determine the conditions of activation for AM. It seems important to confirm the impact of CS on AM by MTT and cytotoxicity assays.
3. It is difficult for me to identify the differences between the first analysis with all the patients and the second one on only the not smoker controls and the smokers. Is it the same patients which has been used or is it a different group of patients or samples? Please clarify this. The results seems to be different at least for the miRNA-20a and -16.
4. In the figure 5, the authors analysed the protein-protein interactions based on the selection of 8 genes (TGF-beta1, etc…). The authors should clarify how they have selected these genes ?
5. In the figure 8, please explain how you have identified the central role of TP53, ATM, and MYC from the first subnetwork and ESR1, BRCA1, and SMAD4 in the second one. For me, this is not evident after the evaluation of the networks presented in this figure. It is very difficult to read the gene names within the circles, particularly in the dark ones. Please change the color of the text!
6. Since the modulation of TGF-beta signaling pathway seems to be strongly related with the modulation of the specific miRNA analysed in this study, the authors should determine in AM from controls the impact of miRNA inhibitors or mimics on this pathway.
Author Response
In this study, the authors evaluated the impact of cigarette smoke (CS) exposure on the expression of some miRNA and the characterization of the specific effect on the transcriptome. The effect of CS exposure was measured by in vitro experiments with AM from patients with different lung diseases (COPD or lung cancer) and by comparing the response among these different categories of patients. Although the data are interesting, the interpretation of the results is altered by the lack of controls and of the integration of some data.
- We thank the reviewer for his comment. We hope that the revision of our manuscript according to the reviewers´ comments and our responses (in blue) has improved our manuscript making it acceptable for publication.
Major comments
- Since previous reports identified the modulation of others miRNA by CS exposure and/or COPD, it seems important to integrate these data at least for the in silico identification of miRNA targets.
- We thank the reviewer for his suggestions. While we acknowledge the importance of incorporating such information for a more comprehensive analysis, our study is focused on a specific scope, and integrating these additional datasets would extend beyond our current objectives. However, we have extended the ‘Introduction’ section reporting miRNA modulated by the CS that have been investigated by other authors. Moreover, we have added information regarding different target genes that have been identified with different bioinformatic approaches.
- The authors evaluated the impact of CS exposure on cell metabolism (by MTT assay) on THP. The decrease in MTT oxidation might be related to cytotoxicity. This might be measured by a specific test. Moreover, this experiment is used in order to determine the conditions of activation for AM. It seems important to confirm the impact of CS on AM by MTT and cytotoxicity assays.
- We thank the reviewer for this remark and recognize the importance of specific tests to confirm the impact of CS on AMs. Considering the extremely precious nature of BAL and the low number of cells that can be obtained from it, we were unable to investigate the cytotoxicity of different CS concentrations for each sample. Therefore, TH-P1 cells were chosen to fill this gap and assess the highest achievable point (in terms of CS percentage treatments), while preserving as many as possible primary cells for miRNA evaluation. However, using the remaining samples during these days, we performed an additional experiment in AMs obtained from three healthy never smokers donors to confirm TH-P1 results (all samples were run 8 times). As can be seen, MTT assay in AMs obtained from three healthy never smokers doners confirmed that the maximum reduction in metabolic activity is reached after 10% CS for 24h in all samples, corroborating what was observed in TH-P1 cells. We have added this information to the manuscript as “data not shown”.
- It is difficult for me to identify the differences between the first analysis with all the patients and the second one on only the not smoker controls and the smokers. Is it the same patients which has been used or is it a different group of patients or samples? Please clarify this. The results seems to be different at least for the miRNA-20a and -16.
- We apologize for confusing the reviewer regarding the CS-focused sub-analysis. We believe that COPD and NSCLC represent two confounding variables which make data interpretation challenging. Moreover, the HS group consisted of 11 subjects, of which 3 were former smokers and 8 were current smokers. Therefore, to highlight the effects of active CS on miRNA expression levels, we performed a sub-analysis focusing on never-smokers (n= 9) in the HNS group exposed or not to in vitro CS, and miRNA expression levels observed in current smokers(n=8) in the HNS group. This analysis allowed us to exclude miRNA modulation by pathological conditions and uncover direct and causal effects of CS. We have edited the manuscript to clarify the analysis criteria.
- In the figure 5, the authors analysed the protein-protein interactions based on the selection of 8 genes (TGF-beta1, etc…). The authors should clarify how they have selected these genes ?
Thank you for your comment. The selection of nodes in Figure 5 was based on their biological relevance and key metrics generated using the MCODE app in Cytoscape software. To clarify this process, we have included the following statement in the Materials and Methods section: "The most relevant nodes were selected based on their biological relevance and metrics such as betweenness centrality, closeness centrality, and topological coefficient." This addition ensures transparency regarding the criteria used for node selection.
- In the figure 8, please explain how you have identified the central role of TP53, ATM, and MYC from the first subnetwork and ESR1, BRCA1, and SMAD4 in the second one. For me, this is not evident after the evaluation of the networks presented in this figure. It is very difficult to read the gene names within the circles, particularly in the dark ones. Please change the color of the text!
We thank the reviewer for his comments. We apologize for any inconvenience caused and have implemented the following changes to enhance the clarity and interpretation of Figure 8. To address the difficulty in reading gene names, particularly within darker circles, we have modified the colour scheme of the network graphs. Specifically, the overall graph colours have been updated to improve contrast and comprehension. In the Materials and Methods section, we have added the following clarification: "The most relevant nodes were selected based on biological relevance, metrics such as betweenness centrality, closeness centrality, and topological coefficient." We have expanded the description in the Results section to highlight how centrality metrics were used to identify these key nodes. The revised text now reads: “CREB1 represented as the central seed node, interacts with multiple highly connected proteins, such as HIF1A, TP53, MYCN, ATM and BCL2 from the first subnetwork and E2F1 (seed node), STAT3, MYC, MAPK1 and BRCA2, from the second network, suggesting that they may play key roles in the networks”.
- Since the modulation of TGF-beta signaling pathway seems to be strongly related with the modulation of the specific miRNA analysed in this study, the authors should determine in AM from controls the impact of miRNA inhibitors or mimics on this pathway.
- We thank the reviewer for this remark and recognize his suggestion as an exciting starting point that could allow us to understand the interplay between CS and immune cell properties. However, the aim of the present study was to elucidate the role of CS in modulating specific miRNAs related to pathological lung conditions. Associating miRNAs with specific pathways is an interesting approach but we believe that this is beyond our focus. Indeed, all chosen mRNAs’ targets were selected from those already validated by several authors. For instance, Kovalchuk et al. assessed the regulation of hsa-miR-17-5p on c-MYC and BCL2 expression using Western blotting. Specifically, the authors reported that miR-17 family regulate proliferation and apoptosis in a model of normal human-derived tracheal/bronchial epithelial cells exposed to ionizing radiation through BCL2 modulation and binding to its 3'-UTR (PMID 20643754). Correia et al. revealed the presence of a match between hsa-miR-20a -5p 3′UTR regions and TGFβ signaling members ALK5, TGFBR2 and SARA trough luciferase reporter assay, emphasizing its significance in the epithelial mesenchymal transition (PMID 26729221). Therefore, our analysis would not add new information to what has already been reported in the literature. We believe it would be interesting to build models of CS exposure and assess miRNA-target correlations, but as there are many pathways related to lung diseases (TGFβ, FoxO, SMAD, or p53), this would imply setting large experiments that, in our opinion, could constitute new projects.

Reviewer 2 Report
Comments and Suggestions for Authors
The manuscript, entitled Epigenetic changes in alveolar macrophages induced by cigarette smoke: miRNA signatures linked to lung disease, assesses the effect of in vitro CS exposure on the expression of several, previously studied in BALF, miRNAs in AMs. Although the study group is relatively small, the results are interesting and novel. I have no major comments on the manuscript, except that In vitro and in vivo should be written in italics. Furthermore, in the Results section, when discussing hsa-miR-223-5p and 20a-5p expression, it is incorrect to refer to Figure 7 (instead of Figure 3).
Author Response
The manuscript, entitled Epigenetic changes in alveolar macrophages induced by cigarette smoke: miRNA signatures linked to lung disease, assesses the effect of in vitro CS exposure on the expression of several, previously studied in BALF, miRNAs in AMs. Although the study group is relatively small, the results are interesting and novel. I have no major comments on the manuscript, except that In vitro and in vivo should be written in italics. Furthermore, in the Results section, when discussing hsa-miR-223-5p and 20a-5p expression, it is incorrect to refer to Figure 7 (instead of Figure 3).
- We would like to thank the reviewer for his observations. Following his suggestions, we have edited the manuscript. We hope that the revision of our manuscript according to the reviewers´ comments are further improving our manuscript.
